

# Real-time task parameter selection method of accounting system based on multi-objective optimization and genetic algorithm

Rongjie Qin[1,2] and Muhammad Shahbaz[3]

[1] Wuhan Technology and Business University, Wuhan, China
[2] Hubei Business Service Development Research Center, Wuhan, China
[3] University of Engineering and Technology, Lahore, Pakistan

## ABSTRACT

The progress of the digital economy has promoted the enterprise accounting system. To accelerate the update and evolution of accounting systems, we propose a parameter selection method based on multi-objective optimization and genetic algorithm. Firstly, this article proposes an accounting feature extraction method based on multimodal information embedding. The dual-branch structure and feature pyramid network are used to realize the feature extraction of the information involved in accounting. Then, this article proposes a multi-objective parameter selection method based on a parallel genetic algorithm. By embedding a genetic algorithm in the process of dual-branch model training, the model's ability to sense accounting information is improved. Finally, using the above two methods, an accounting system evaluation method upon recurrent Transformer is proposed to improve the financial situation of enterprises. Our experiments have proven that our approach attains a remarkable performance with an 87.6% F-value, 83.5% mAP value, and 83.4% accuracy. These results position our method at an advanced level globally, showcasing its capability to enhance accounting systems.

# INTRODUCTION

The digital economy era has brought many new business models, forms of data, and ways of processing information. In addition, the era of the digital economy has produced massive amounts of data, which urgently needs to be managed intelligently. Research on a brand-new accounting system helps to formulate standards that adapt to new business models, can help design efficient data processing methods, make the extraction, sorting and analysis of accounting information more rapid and accurate, and boost the credibility and transparency of finance.

Through the study of accounting algorithms, the existing accounting theory system can be gradually improved (*Dwirandra & Astika, 2020*). New algorithms may involve more

Corresponding author
Rongjie Qin, qinrongjie@wtbu.edu.cn

complex business models, data structures and information processing methods to promote the development of accounting theory and make it better adapted to the needs of the digital economy era. Research on accounting algorithms helps boost accuracy, reliability and timeliness and can optimize the process of financial information collection, processing and reporting, thereby improving the quality of financial information (*Xolmirzaev, Juraev & Axmadjonova, 2021*). In addition, in the environment of the digital economy, the research of accounting algorithms can help to expand the theory of risk management, including how to better identify, assess and deal with new risks in the digital economy. At the same time, the research of accounting algorithms can also provide the foundation of the intelligent decision support theory. Using algorithms and artificial intelligence technology, financial data can be analyzed more comprehensively and accurately (*Miller & O'leary, 2019*). However, the research of accounting algorithms involves the complexity and diversity of the financial field, and there are the following difficulties. First of all, different industries and enterprises have different business models and modes of operation, which require customized accounting algorithms, and the algorithms with strong versatility cannot cover all situations (*Salam, 2022*; *Gofwan, 2022*; *Gao, 2022*). Then, accounting involves changes in standards and a lot of uncertainty factors. Therefore, it's necessary to ensure the applicability and compliance of the algorithm (*Asiaei et al., 2022*; *Han et al., 2023*). Solving these difficulties requires interdisciplinary collaboration, combining knowledge from fields such as accounting, information technology, and data science.

Using artificial intelligence and big data technologies, research on accounting systems can provide more intelligent financial decision support for enterprises. Through data analysis and prediction models, enterprises can better understand their financial situation and make more informed strategic decisions. *Zhou (2023)* proposed two accounting management prediction models. By integrating CNN and GRU to construct two different models, the accounting management index system is established, which exerts a guiding influence on the economic advancement of enterprises. *Li, Khishe & Qian (2023)* proposed a model for financial accounting information systems to solve the problem due to the increasing size of the data set. The system uses five DGRU-based models to improve the accuracy of profit forecasting. *Ionescu (2020)* analyzed algorithmic data-driven accounting systems to analyze and estimate the utilization of AI in the fields of audit, tax, and management accounting. *Li (2022)* introduced a parallel bookkeeping approach utilizing deep neural networks. By exploring the parallel bookkeeping pathway, this approach enhances the efficiency and precision of financial personnel. By creating an intelligent recognition model, *Feng & Zhong (2022)* addressed the limitations of conventional accounting methods, including significant calculation errors, time intensiveness, and labor intensiveness. Their work serves as a crucial reference for enterprises in making significant decisions. *Chowdhury (2023)* proposed a management information prediction by neural network, which encompasses five dimensions. *Ushio & Yamamoto (2021)* constructed a deep learning algorithm model upon a snapshot of time series financial data to identify accounting information of growing companies.

However, the above methods do not take into account the multiple forms of features contained in the accounting system and cannot realize the comprehensive judgment

of the whole system. In addition, the accounting system also needs to have high real-time performance. Therefore, we propose a real-time method for parameter selection by the multi-objective optimization (MOO) and genetic algorithm (GA) to realize the optimization of the accounting system. The main contributions are as follows:

1. We propose a method for extracting accounting features by incorporating multimodal information embedding, aiming to enhance the overall quality of enterprise accounting.

2. We suggest an approach for selecting parameters using a parallel genetic algorithm with a multi-objective perspective to improve performance and adaptability in diverse scenarios.

3. We introduce a method for evaluating accounting systems that utilize a loop Transformer to achieve precise modeling of enterprise accounting systems and fulfill the requirements of system implementation accurately.

## RELATED WORKS

### Accounting techniques

Accounting systems are widely used at home and abroad. The United States General Group uses the UNIVACI system, and it only takes four hours to calculate and print the wages of thousands of people. Since then, the use of computers for auxiliary management in enterprises has entered a stage of rapid development.

*Zhang, Wang & Dong (2018)* and *Verhoef et al. (2021)* invented and implemented the accounting automation robot and the corresponding business process automation system. The employment of intelligent information technology in corporate accounting will become more and more in-depth. Companies can not only use automated robots to complete the interaction between various information systems but also independently review the risks in financial management, evaluate the soundness of the company's financial practices, and protect the company's property safety to the maximum extent through the comprehensive and systematic management of financial data. Improve the quality of information. *Zhang & Zhang (2018)* believe that in the competitive external market environment, managers need to make accurate business decisions to maintain the competitive advantage of enterprises, and the application of accounting systems can enable enterprises to quickly find business problems and help management to make effective management decisions. *Duan (2023)* analyzed the advantages and functions of cloud computing platforms, and then expounded on the development status and existing problems of intelligent accounting, and finally put forward some application measures of cloud computing platforms in financial management, hoping to provide some useful reference for relevant enterprise financial personnel, to effectively promote the intelligent development of enterprise financial management. *Wang (2023)* take the intelligent construction of enterprise accounting as the analysis object. Firstly, it summarizes the content and characteristics of the intelligent construction of enterprise financial accounting, then analyzes the existing problems in the intelligent construction of enterprise accounting, and finally discusses the countermeasures of the intelligent construction of enterprise accounting system and does the basic work of financial intelligence. To boost the financial intelligence construction and promote

the sound and rapid development of enterprises. *Petri-şor & Cozmiuc (2016)* suggested that intelligent accounting shared services could be overseen as a collection of varied enterprise activities encompassing straightforward iterative processes. *Cooper et al. 2019*) built intelligent accounting analysis and diagnosis robots to evaluate the comprehensive financial performance of listed companies. *Lai (2022)* took telecom enterprises as an example, focusing on the influence of enterprise intelligent accounting systems on the communication industry based on the perspective of artificial intelligence, covering the two-layer architecture of the underlying support system and the upper intelligent financial application.

### Intelligent accounting technology

By applying advanced technologies to improve and automate the accounting process, the application of these technologies can help enterprises improve the efficiency of accounting, reduce errors, and enhance the accuracy and credibility of accounting data.

*Wang et al. (2020)* believe that accounting services based on deep neural networks can optimize accounting processes and centralize financial functions to boost the transformation of financial personnel and enhance the ability of enterprises to quickly respond to business needs and the efficiency of financial management, to maximize the value of the company. *Ghimire et al. (2020)* proposed intelligent accounting services based on big data and artificial intelligence to realize new financial and tax models such as automatic bill identification, semantic parsing, automatic bookkeeping, machine audit, automatic reconciliation and even one-click tax return, which greatly improves the efficiency of financial and tax work and reduces human errors. Financial risk identification is realized through modeling. On the macro level, *Huang, Qiu & Li (2021)* prejudged and prevented the financial risks of the real economy by establishing a comprehensive assessment and early warning of short-term financial risks considering both stock and flow. On the micro level, *Goodell et al. (2021)* constructed an intelligent accounting risk identification system applying machine learning based on word frequency statistics technology. It provides a reference for the construction of a data factor market and the implementation of an artificial intelligence strategy. *Uthayakumar et al. (2020)* also introduced a classification model utilizing enhanced K-means with fitness consideration for predicting the accounting system and monitoring the timely financial performance of enterprises.

The intelligent development of accounting systems encompasses various aspects, such as the use of automated robots, business process automation systems, cloud computing platforms, and artificial intelligence (AI) technology applications. In terms of enterprise accounting construction, through intelligent accounting sharing services, accounting analysis, and diagnostic robots, among other methods, enterprises can efficiently manage diverse activities and assess overall financial performance. Regarding intelligent accounting technology, the application scope of AI includes deep neural networks, big data, and machine learning, providing robust support for optimizing accounting processes and consolidating financial functions. The intelligent evolution of accounting systems presents both opportunities and challenges for enterprises. By consistently introducing advanced technology and refining management approaches, businesses can adapt to the competitive

environment, enhance managerial efficiency, and achieve sustainable development. Nevertheless, as technology continues to progress, it is crucial to closely monitor potential issues and risks to ensure the effective and secure implementation of intelligent applications in enterprises.

## Real-time task parameter selection method based on multi-objective optimization and genetic algorithm

Considering that the multi-form features contained in accounting make it difficult to characterize the whole system and improve the real-time performance and accuracy of the accounting system, we propose a real-time task parameter selection method by multi-objective optimization and genetic algorithm to optimize the financial situation of companies. To improve the quality of the enterprise accounting system, this article proposes an accounting feature extraction method based on multimodal information embedding, a multi-objective parameter selection method based on a parallel genetic algorithm, and an accounting system evaluation method based on a loop Transformer in turn, to realize the accurate modelling of the enterprise accounting system and complete the requirements of system implementation.

The flow chart of the whole method is in Fig. 1. It can be seen that they respectively start from the perspectives of multimodal information synthesis, multi-objective optimization and time series modelling, providing a more comprehensive, flexible and accurate means to improve the enterprise accounting system. Firstly, the accounting feature extraction method based on multimodal information embedding can involve different types of data, such as text, images, time series, etc., to describe the financial situation of enterprises more comprehensively. By integrating different modalities of information, richer and more comprehensive features can be extracted, and the system can better understand and analyze the financial data of enterprises. This is helpful to improve the prediction accuracy and anomaly detection ability, thus enhancing the quality of the accounting system. Secondly, multi-objective parameter selection based on a parallel genetic algorithm can help to find a balance between multiple objectives instead of focusing only on a single objective. By employing the parameter selection facilitated by the parallel genetic algorithm, the system gains enhanced adaptability to diverse business scenarios and requirements. The optimization of multiple objectives contributes to achieving superior performance across various facets of the system, thereby elevating overall quality. Ultimately, the evaluation method for accounting systems based on recurrent transformers enables more precise modeling and prediction of time series data, encompassing financial trends and seasonal variations. This heightened precision in forecasting aids in providing more reliable financial decision support for enterprises, thus contributing to an overall improvement in the quality of accounting systems.

## Accounting feature extraction Method based on multimodal information embedding

According to the characteristics of enterprise accounting information, an accounting feature extraction method based on multimodal information embedding (MIE) was proposed.

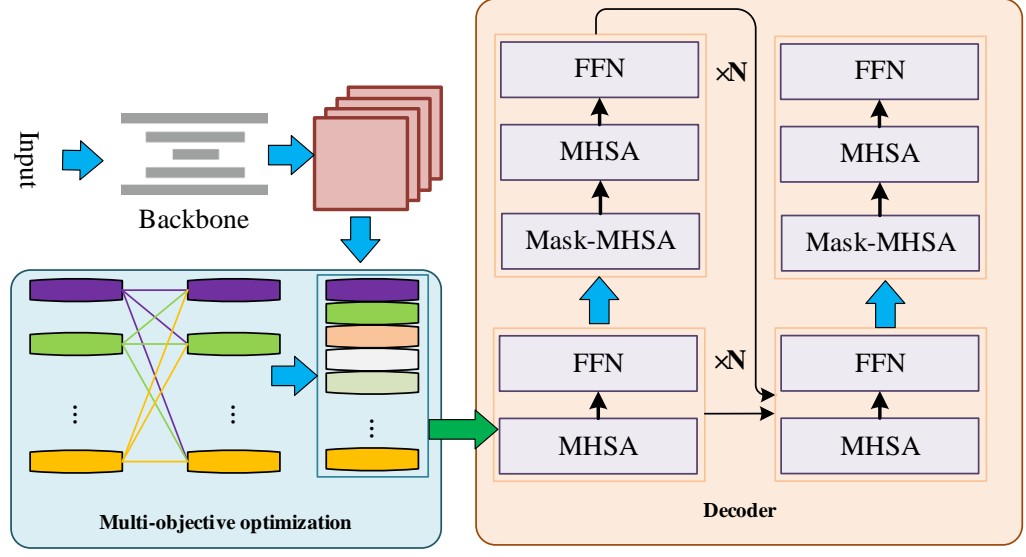

**Figure 1   Global algorithm model.**

The backbone network uses ResNet and FPN pyramid structure as the image feature extractor, and its basic modules are shown in Fig. 2. ResNet is easier to optimize than ordinary networks; The accuracy can be improved from the increase of depth, and its recognition results are excellent. It is highly scalable, and there is still no degradation problem when the network increases to 100 or even 1,000 layers. A parallel sub-network structure is designed after the backbone network, and the features extracted from the backbone network are input into the two module branches as a shared feature layer. The multi-objective optimization subtask and the comprehensive evaluation subtask of the accounting system are trained synchronously to jointly optimize the backbone feature extraction network. Through the bilinear branch, an implicit connection relationship is established between modules, and the correlation between tasks is used to guide the feature extractor to guide the model to solve the optimal parameter selection.

The parallel subnet structure enables the concurrent handling of numerous tasks, leading to an enhancement in overall performance. Each subnet can function independently, eliminating the need to await the completion of other subnets. This accelerates both data transmission and processing speeds. The parallel subnet structure facilitates easy scalability, allowing for flexible adjustments to the system size based on specific requirements, ultimately achieving superior scalability. This architecture excels in simultaneously managing multiple tasks, particularly in scenarios involving intricate computations and extensive data processing, thereby boosting the system's multitasking processing capability.

In the bilinear branch, each layer of the subnetwork is a parallel structure, including accounting multi-objective parameter optimization and classification branches. The multi-objective parameter optimization has the same structure as the classification branch, which shares the input feature layer. The dense sampling strategy is used to establish a

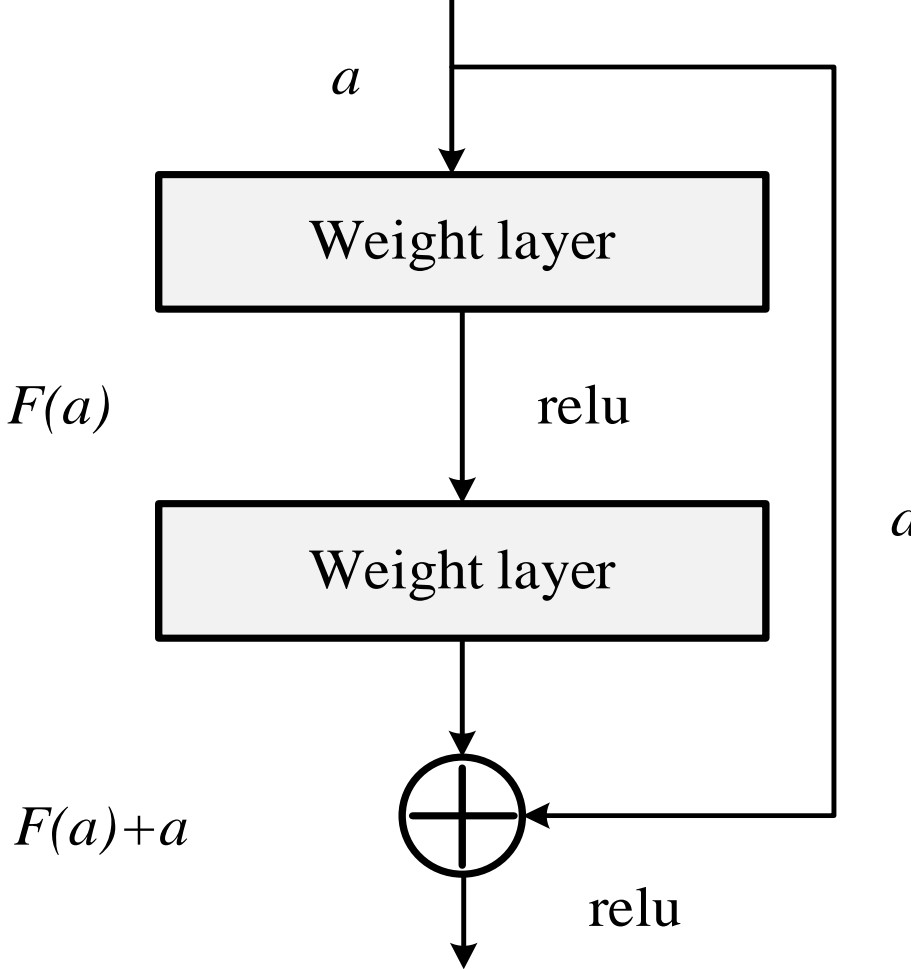

**Figure 2** **Element of our backbone.**

fully connected structure with shared weights on the feature pyramid, and the features at all levels on the shared feature layer are filtered by convolution. The features of the input accounting multi-objective parameter optimization and classification branch are X and Y, respectively. The calculation formula is as follows:

$$X = \sum_{i=1}^{4} f_1[Conv_{2\_x}[Conv_{1\_x}(F)]] \tag{1}$$

$$Y = \sum_{i=1}^{4} f_2[Conv_{3\_x}[Conv_{1\_x}(F)]] \tag{2}$$

where Conv2_x1_x represents a convolutional layer consisting of four 3 × 3 convolutions with the same number of channels, Conv3_x represents a 3 × 3 kernel with nine channels, and Conv represents a 3 × 3 kernel with 36 channels. Multi-objective parameter

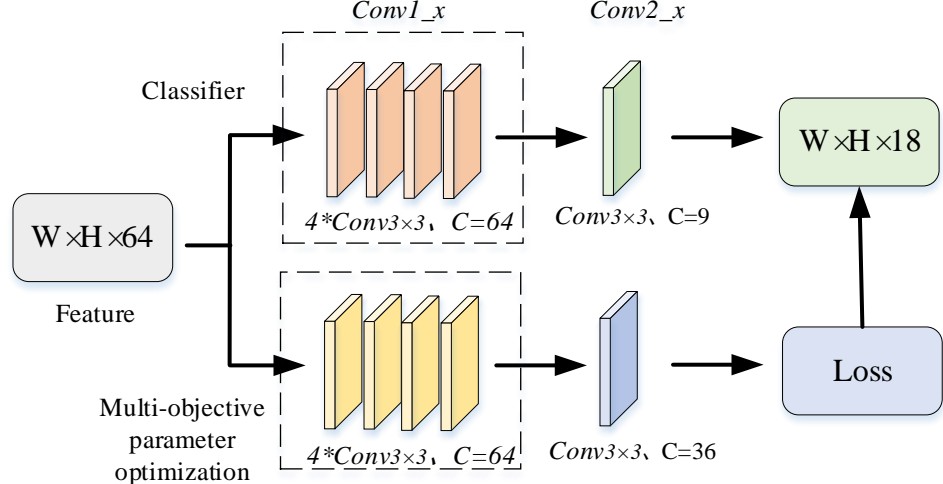

**Figure 3  Structure of parallel structure.**

optimization maps the feature representation space to the target category space by dimension transformation f1.

## Multi-objective parameter selection based on parallel genetic algorithm

Because the enterprise accounting data needs to rely on a large number of characteristic parameters, usually not all of the characteristics of the final system evaluation have a positive impact. To address this issue, we propose a multi-objective parameter selection based on a parallel genetic algorithm (MOPS). The Loss function in genetic algorithm and model training is coordinated to eliminate the influence of redundant features on the model effect.

Firstly, in Fig. 3, the parameter data of the input multi-objective branch are initialized by binary coding, and each feature of each data is selected in {0,1} according to equal probability. The matrix is expressed as follows.

$$\begin{bmatrix} a_{1,x_1} & \cdots & a_{1,x_n} \\ \vdots & & \vdots \\ a_{m,x_1} & \cdots & a_{m,x_n} \end{bmatrix} \tag{3}$$

where $x_j(j = 1, 2, \ldots, n)$ represents the data feature, a value of 0 indicates non-selection, a value of 1 indicates selection, and m represents the length of the time series data. Binary encoding serves as a prevalent representation for problems featuring discrete decision variables. It adeptly maps decision variables to individuals in genetic algorithms. Binary encoding offers flexibility in representing different decision variables based on problem structures, enabling genetic algorithms to better align with problem characteristics.

Secondly, this article chooses the separability judgment basis as the criteria for the fitness function. Suppose D optimal features are selected from d features, which are defined as

follows.

$$S_b = \sum_{i=1}^{c} P_i(\mathbf{h}_i - \mathbf{h})(\mathbf{h}_i - \mathbf{h})^{\mathrm{T}} \qquad (4)$$

$$S|_w = \sum_{i=1}^{c} P_i \frac{1}{q_i} \sum_{k=1}^{q_i} \left(x_k^{(i)} - \mathbf{h}_i\right)\left(x_k^{(i)} - \mathbf{h}_i\right)^{\mathrm{T}}, \qquad (5)$$

$$J_d(x) = \frac{\mathrm{tr}\mathbf{S}_b}{\mathrm{tr}\mathbf{S}_w} \qquad (6)$$

In the provided formula, hi denotes the mean of the sample set across all classes, Pi is the prior probability of class i, Sb represents the between-class matrix, Sw is the within-class matrix, and trSw and trS denote the trace of the two matrices.

Finally, after determining the fitness function, the genetic algorithm evaluates the individual based on the fitness value. In this investigation, the selection operator employed is the roulette wheel method. In the crossover operator, multiple crossover points are randomly set in a single individual, and then two individuals are exchanged for genes according to these crossover points to generate new individuals. In addition, single-point position flip mutation is used as the mutation operator in this article. In this method, an individual is randomly selected from the population, and then a gene of the individual is randomly selected for flipping mutation.

Through the selection of a genetic algorithm, the constraint process of training loss function can be expressed as follows.

$$L_a^i(\partial_{f,r,s}) = L_a([[x_i, \partial_f], \partial_x], X) \qquad (7)$$

$$L_t^i(\partial_{f,r,s}) = L_t([[x_i, \partial_f], \partial_y], Y) \qquad (8)$$

$$G(\partial_{f,r,s}) = \frac{\delta}{n} \sum_{i=1}^{n} L_a^i(\partial_{f,r,s}) + \frac{n-\delta}{n} \sum_{i=1}^{n} L_t^i(\partial_{f,r,s}) \qquad (9)$$

where G(.) Represents the model optimization objective.

## Evaluation method of accounting system based on Loop Transformer

The evaluation method of accounting systems based on Loop Transformers (CT) utilizes loop transformers as the main tool to evaluate and analyze the performance of accounting systems more comprehensively and dynamically.

Firstly, we discard the encoder in Transformer, which is similar to BERT, but its input structure is different. In this article, for any feature, I, the extracted features F and X are used to construct the backbone network of the model containing only the Transformer layer, as shown in Fig. 4. Each Transformer layer contains two multi-head attention (MHSA), layer

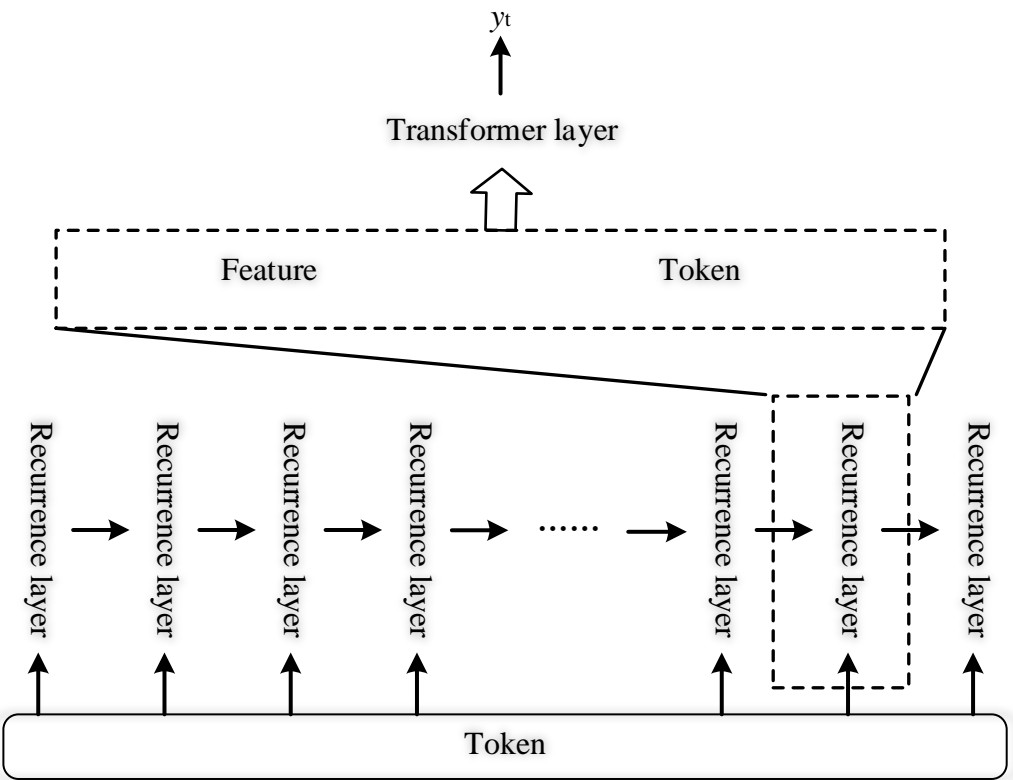

**Figure 4** Structure of recurrent transformer.

normalization (LayerNorm), FeedForward layer (FFN), and residual connection structure. Defining a single Transformer layer as T, the computation of layer i is as the following formula:

$$
\begin{aligned}
X_{D_i} &= T(X, F) \\
&= T(MHSA(X, X, X), LN, MHSA(F, X, X), LN, FFN)
\end{aligned}
\tag{10}
$$

where $i \in N$, N refers to the number of layers.

As the input of the recurrent layer, which contains the mutual relationship, the recurrent layer acts as the core of the structure, thereby enhancing the depth of the Transformer model. The recurrent layer is Rt and the structure is shown in Fig. 4. By concatenating these features and feeding them into the Transformer layer, the classification result can be directly obtained, and the process is shown in the following formula:

$$
F' = \left[ W_f F, W_x X \right]
\tag{11}
$$

$$
h_t = T\left( F', F' \right)
\tag{12}
$$

$$
y_t = \text{Softmax}(W_h h_t)
\tag{13}
$$

**Table 1  Implementation parameters.**

| Parameters | Value |
|---|---|
| Initial learning rate | $2 \times 10^{-4}$ |
| Epoch | 45 |
| Batch-size | 32 |
| Decay | 0.94 |
| Gradient descent method | SGD |
| Deep learning framework | Tensorflow |
| CPU | i7-13700 h |
| GPU | 2*Nvidia RTX4070 |

where $W_f, W_x \in \mathbb{R}^D$ represents the learnable linear transformation matrix, [,] represents the concatenation operation for features, T denotes the operation of the Transformer layer, and Softmax refers to the activation function to obtain the evaluation result.

# EXPERIMENT AND ANALYSIS

## Dataset and implement details

We use Synthetic Financial Dataset (https://zenodo.org/records/7543591, 10.5281/zenodo.7543591) to test the real-time tasks based on multi-objective optimization and genetic algorithm parameter selection method. In this article, the configuration of i7-13700 h and 2*Nvidia RTX4070 is used to complete the training of the method. The deep learning protocol we used is Tensorflow, and the adopted training Settings are shown in Table 1. The model is trained with a weight decay term of 0.008. At the same time, we use SGD as the optimizer of the model. The processing of accounting data encompasses several stages, commencing with the collection of financial transactions and information from diverse sources, including sales, procurement, payments, receipts, etc. Subsequently, the data undergoes classification and encoding and is then transformed into a format suitable for financial reporting through the application of various accounting rules and principles. This process involves tasks such as computing the general ledger, crafting financial statements, and calculating profit and loss, among others.

Since the accounting process designs a variety of different forms of characteristics, we use mean square error (mAP), F-measure (Z) and accuracy (Acc) as the evaluation criteria of the method in this article, and the formulas are as follows:

$$Y_{Pre} = \frac{V(gt \cap pr)}{V(pr)} \tag{14}$$

$$X_{Re} = \frac{V(gt \cap pr)}{V(gt)} \tag{15}$$

$$Z_F = \frac{2 \times Y_{Pre} \times X_{Re}}{Y_{Pre} \times X_{Re}} \tag{16}$$

$$mAP = \frac{1}{N} \sum Y_{Pre} \times X_{Re} \tag{17}$$

$$Acc = \frac{V(pr)}{V(gt)} \tag{18}$$

In these formulas, pr and gt denote the predicted results and true values of the model, respectively, while V(.) represents the statistical value. Re signifies recall, and Pre corresponds to precision. With the above training Settings and evaluation indicators, we test the performance of the accounting feature extraction method based on multimodal information embedding, the multi-objective parameter selection method based on parallel genetic algorithm, and the accounting system evaluation method based on recurrent Transformer. In addition, for the whole accounting system, we use the model calculation amount, model parameter number, and operation time to evaluate the performance.

## Ablation experiments

We conduct ablation experiments of three sub-methods on the Synthetic Financial Dataset. From Fig. 5 and Table 2, it can be found that the baseline we adopted is a Transformer, which can obtain a 78.6% F value and 77.3% mAP value. Firstly, we embed MIE, MPOS, and CT modules on the basis of baseline and can obtain 81.3%, 82.6%, and 82.2% F values, respectively, which proves the performance of the above three modules and can strengthen the baseline. Then, on the basis of MIE, MPOS and CT, we perform pairwise combination embedding respectively. MIE&MPOS can achieve 85.3% F value and 81.4% mAP value, MIE&CT can achieve 85.7% F value and 82.2% mAP value, and MPOS&CT can achieve 84.8% F value and 82.0% mAP value. Finally, three modules embedded in the baseline simultaneously can obtain an 87.6% F value and 83.5% mAP value. The above results show the performance of the model under different module combinations. By comparing the combinations of different modules, we can see the improvement effect of each module on the model performance, and the synergistic effect between different combinations of modules. In the end, the best model performance is achieved by embedding all three modules into the baseline.

## Compare our method and other methods

Firstly, we conduct the performance experiments of the accounting feature extraction method based on multimodal information embedding on the Synthetic Financial Dataset. We selected CNN (*Kattenborn et al., 2021*), LSTM (*Yu et al., 2019*), Transformer (*Vaswani et al., 2017*), BERT (*Deepa, 2021*) and TypeFormer (*Chen et al., 2022*) to compare the performance in Fig. 6 and Table 3. We can find that MIE achieves the highest value in all evaluation indicators, with recall, precision and F-measure scores of 0.835, 0.867 and 0.844, respectively. Compared with CNN and LSTM alone, our method improves the F-score of the model by more than 8%, mainly because CNN and LSTM are difficult to deal with multiple morphological features and are far less compatible and scalable than MIE when dealing

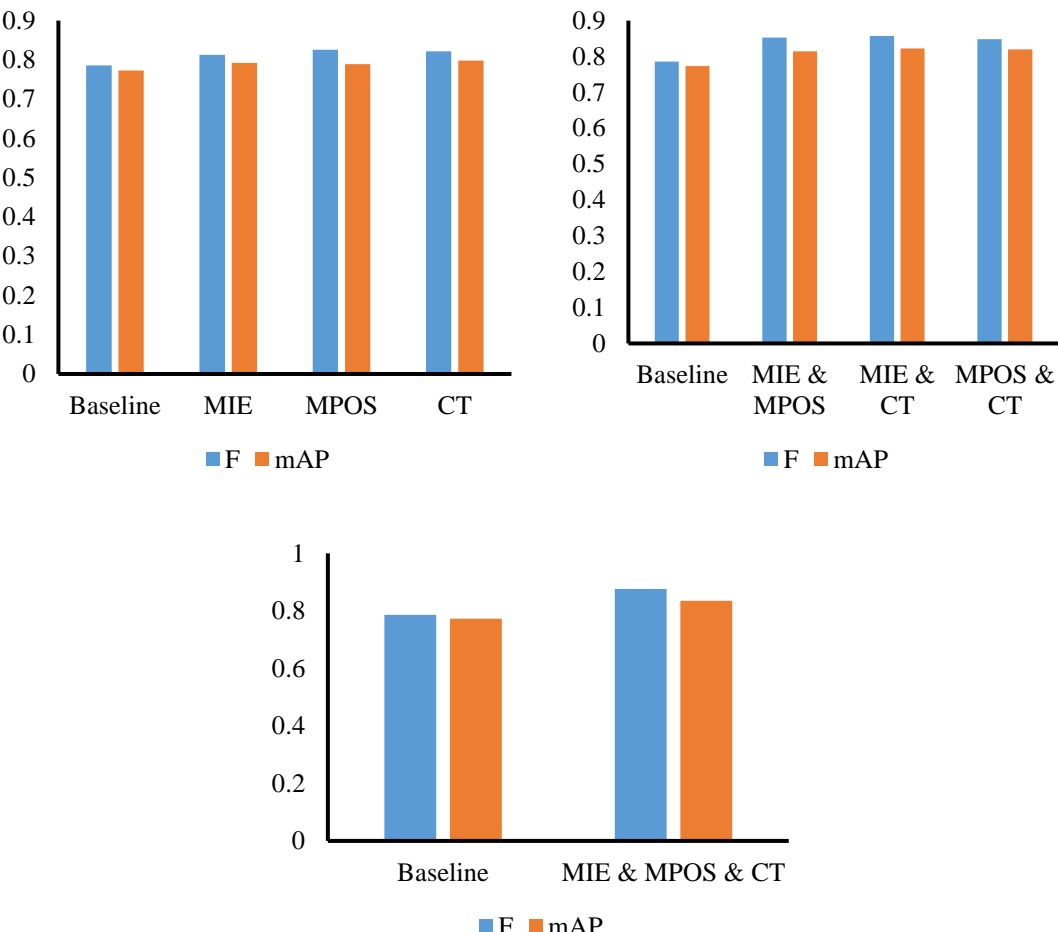

**Figure 5  Ablation experiments results.**

with different semantic features. Compared with Transformer and BERT, our method has an F-score lead of 2%. Transformer and BERT have excellent multimodal processing ability. However, this article further improves the feature representation ability through the two-branch structure, which has good performance. Compared with TypeFormer, MIE still has advantages in terms of F-measure, precision and recall. For the feature extraction of accounting systems, the accounting feature extraction method by multimodal information embedding has a great lead. Through the feature extraction backbone network and the dual-branch structure, it can effectively extract and align various features in accounting.

Then, we test the performance of multi-objective parameter selection based on a parallel genetic algorithm on the dataset. MPOS will still be compared with CNN (*Kattenborn et al., 2021*), LSTM (*Yu et al., 2019*), Transformer (*Vaswani et al., 2017*), BERT (*Deepa, 2021*), and TypeFormer (*Chen et al., 2022*). In this experiment, the model will be evaluated in terms of accuracy, number of parameters, and elapsed time. It can be found from Figs. 7 and 8 and Table 4 that MPOS achieves an accuracy score of 0.823, an inference time of 89 ms, and a model parameter number of 512, which reaches the world advanced level.

| MIE | MPOS | CT | F | mAP |
|---|---|---|---|---|
| | | | 0.786 | 0.773 |
| O | | | 0.813 | 0.792 |
| | O | | 0.826 | 0.789 |
| | | O | 0.822 | 0.798 |
| O | O | | 0.853 | 0.814 |
| O | | O | 0.857 | 0.822 |
| | O | O | 0.848 | 0.820 |
| O | O | O | 0.876 | 0.835 |

**Table 2  Ablation experiments.**

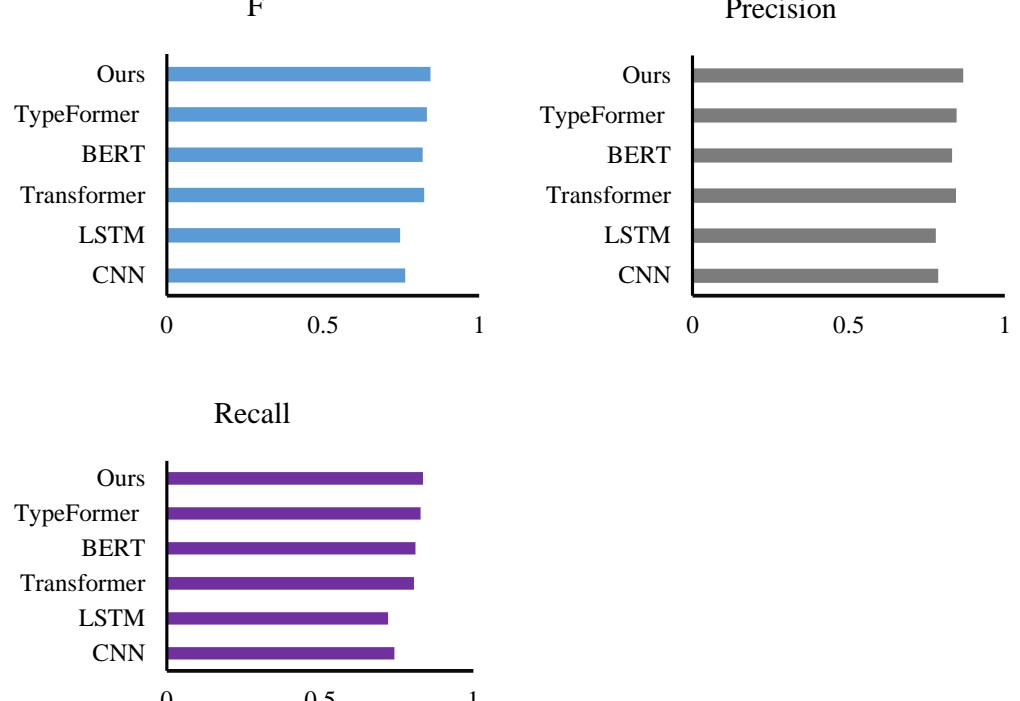

**Figure 6  Comparison with other extraction methods.**

Compared with Transformer, BERT, and TypeFormer, MPOS has higher model accuracy, shorter inference time, and smaller model parameter numbers. As a training method optimized by genetic algorithm, MPOS adjusts the two-branch CNN method to make it compatible with multimodal accounting features. Compared with CNN and LSTM, MPOS has a model accuracy lead of more than 8%. Due to the addition of the training method and genetic algorithm, the inference time and the number of parameters of the model are

**Table 3  PRF values for different model methods.**

| Methods | F | Precision | Recall |
|---|---|---|---|
| CNN | 0.763 | 0.787 | 0.742 |
| LSTM | 0.747 | 0.779 | 0.722 |
| Transformer | 0.824 | 0.844 | 0.806 |
| BERT | 0.819 | 0.831 | 0.811 |
| TypeFormer | 0.833 | 0.846 | 0.828 |
| Ours | 0.844 | 0.867 | 0.835 |

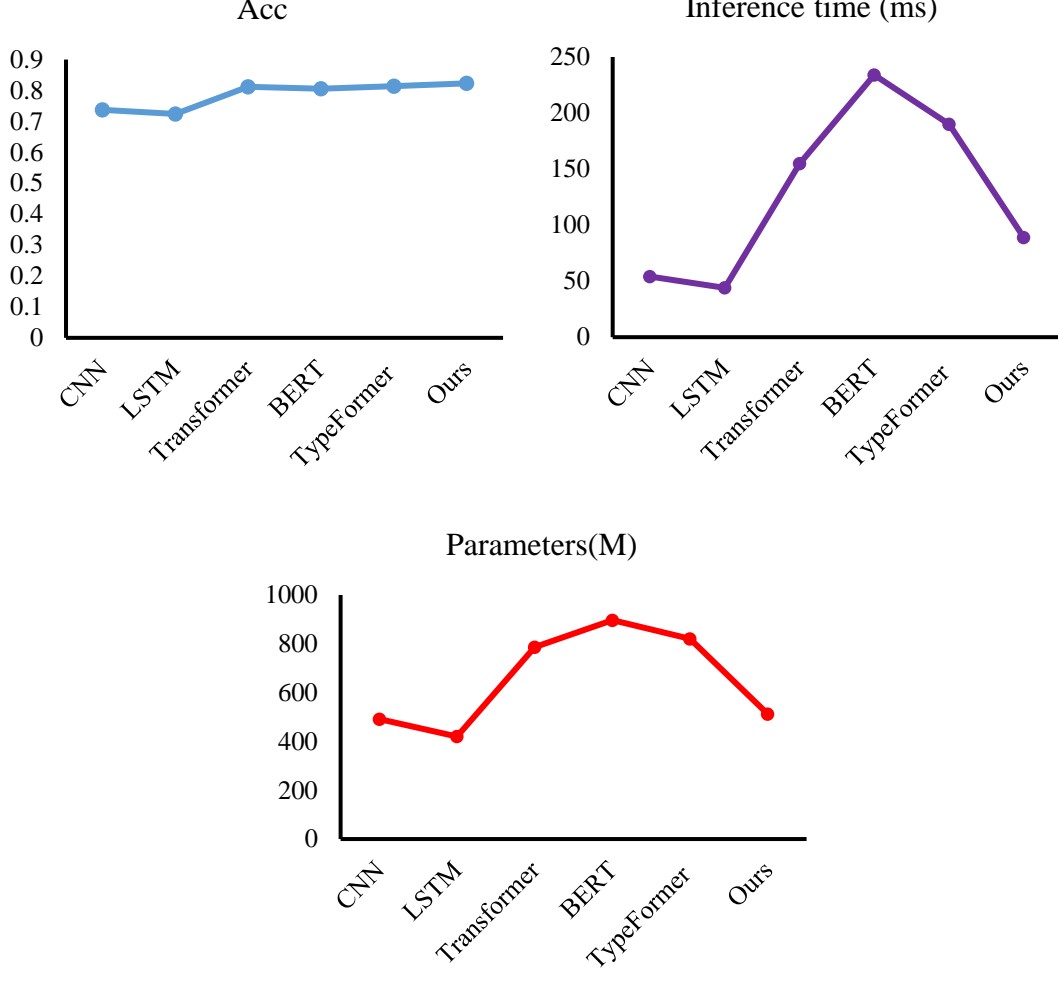

**Figure 7  The results of our method.**

increased. However, MPOS achieves great performance improvement with a small resource consumption.

Finally, we verify the performance of the cyclic transformer-based evaluation method for accounting systems. By comparing M2-Transformer (*Cornia et al., 2020*), LSTNet

| Confuse Matrix | | Predict | |
|---|---|---|---|
| | | True | False |
| GT | True | 235 | 16 |
| | False | 12 | 219 |

**Figure 8  Confuse matrix for MPOS.**

**Table 4  Compare MPOS with other methods.**

| Methods | Acc | Inference time (ms) | Parameters(M) |
|---|---|---|---|
| CNN | 0.737 | 54 | 492 |
| LSTM | 0.724 | 44 | 420 |
| Transformer | 0.812 | 155 | 786 |
| BERT | 0.806 | 234 | 897 |
| TypeFormer | 0.814 | 190 | 821 |
| Ours | 0.823 | 89 | 512 |

(*Li et al., 2020a*), OSCAR (*Li et al., 2020b*), and VinVL (*Zhang et al., 2021*), our method obtains 87.6% F-score, 83.5% mAP value, and 83.4% accuracy, and the results are shown in Fig. 9. In this article, the F-measure of the model is boosted by at least 2.0%, and the accuracy value is improved by at least 1.3%. In the test, MIE and MPOS are embedded into CT at the same time, and a complete accounting system evaluation method is obtained. Under the effect of the dual-branch structure and the loop structure, the proposed method achieves the highest evaluation index, which not only exceeds various variants of the Transformer but also exceeds the model trained on big data, proving the effectiveness and scalability of the proposed method.

Considering the real-time nature of the accounting system, this study adopts a single model. A single model is usually simpler and easier to interpret. The training and reasoning process of a single model is usually more efficient than that of a mixed model. When resources are limited or real-time performance is required, a single model may be more appropriate. In addition, accounting data has obvious distribution characteristics, patterns or relationships, and a single model can well capture these characteristics, so

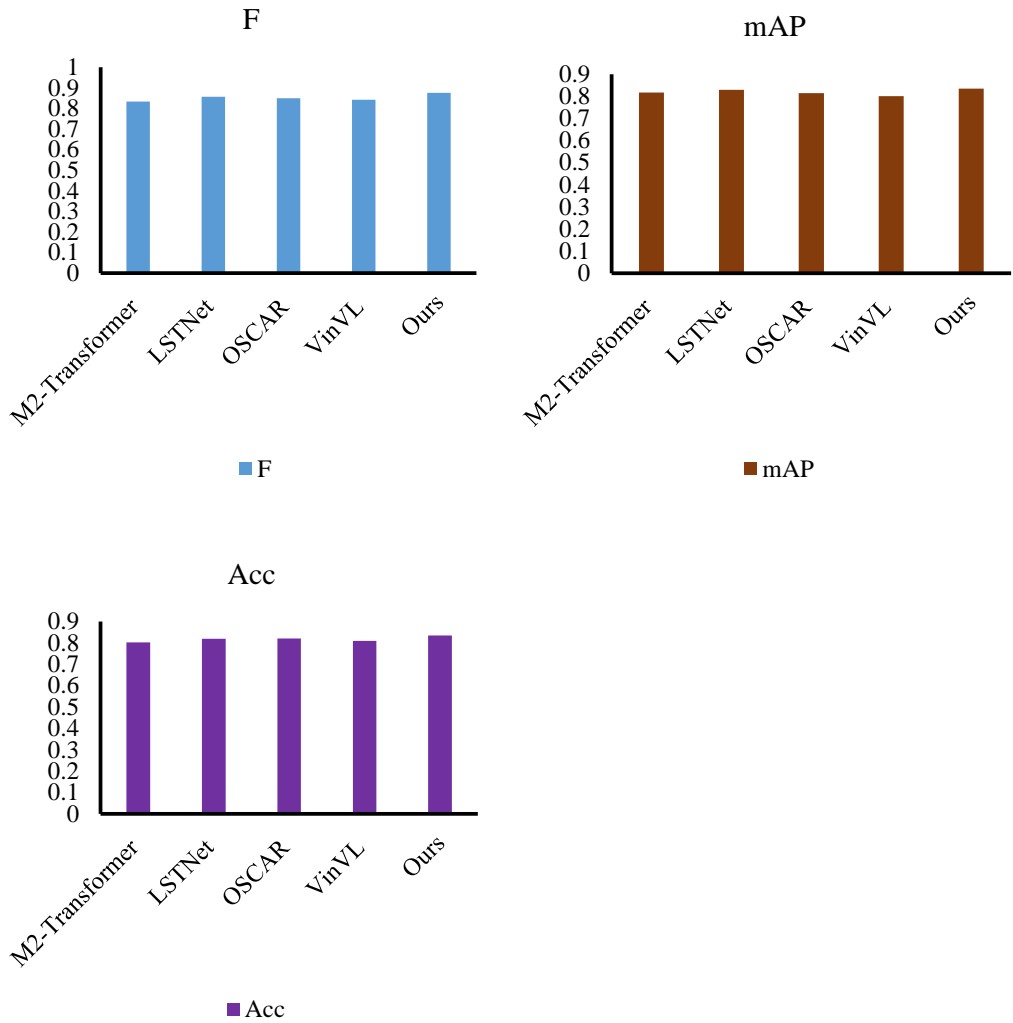

**Figure 9** Model efficiency comparison with other methods.

the introduction of hybrid models may not significantly improve performance, but may increase complexity.

## Discussion

In the experiment to validate performance, we meticulously tested and evaluated the three proposed methods. Initially, leveraging multimodal information embedding enabled the successful extraction of diverse accounting features. Experimental outcomes underscore the pronounced advantages of this method over traditional approaches in capturing features from varied information sources. Comparative analysis revealed a significant enhancement in feature expression capabilities. Subsequently, we employed a parallel genetic algorithm for multi-objective parameter selection to optimize the accounting system's performance. Results demonstrated that this approach comprehensively explores parameter space, identifying optimal parameter sets that strike a balance between multiple objectives.

In comparison to a single genetic algorithm, the parallel method exhibited superior performance, showcasing enhanced convergence speed and global search capabilities. Finally, introducing an evaluation method for accounting systems based on the Circular Transformer involving temporal modeling of accounting data yielded promising results. This method demonstrated heightened accuracy in capturing temporal correlations and trends within accounting systems, offering a robust tool for dynamically assessing system performance (*Beuselinck et al., 2023*).

In conclusion, the performance validation experiments affirm the effectiveness of these three methods in enhancing accounting system efficiency, refining parameter selection, and modelling time series data (*Baker et al., 2023*). These findings provide substantial theoretical and empirical backing for advancing accounting intelligence and system optimization (*Al-Hattami & Kabra, 2024*). In future endeavours, we plan to delve deeper into the practical application areas of these methods and evaluate their performance in real-world scenarios.

## CONCLUSION

To expedite the development of the digital economy for enterprises, we present a parameter selection approach employing multi-objective optimization and genetic algorithms. This method involves introducing an accounting feature extraction technique utilizing multimodal information embedding and a multi-objective parameter selection method based on a parallel genetic algorithm. These approaches collectively address feature extraction, representation optimization, and parameter selection within the accounting process. Leveraging these methodologies, we introduce an accounting system evaluation method using a loop Transformer aimed at enhancing the financial position of enterprises. Our proposed method demonstrates robust model performance and scalability, evident in the F value of 87.6%, mAP value of 83.5%, and accuracy of 83.4%, substantiating its capability to optimize the accounting system.

### Funding
The authors received no funding for this work.

### Competing Interests
The authors declare there are no competing interests.

### Author Contributions
- Rongjie Qin conceived and designed the experiments, analyzed the data, prepared figures and/or tables, authored or reviewed drafts of the article, and approved the final draft.
- Muhammad Shahbaz performed the experiments, performed the computation work, authored or reviewed drafts of the article, and approved the final draft.

## Data Availability

The raw data is available at Zenodo: Open Access. (2023). Synthetic Financial Datasets [Data set]. Zenodo. https://doi.org/10.5281/zenodo.7543591.

## Supplemental Information

Supplemental information for this article can be found online at http://dx.doi.org/10.7717/peerj-cs.1952#supplemental-information.

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
