# Peer review of "Real-time task parameter selection method of accounting system based on multi-objective optimization and genetic algorithm"

_PeerJ Computer Science, doi:10.7717/peerj-cs.1952_

## Round 0.1 · original submission · Major Revisions

The feedback from both reviewers addresses several key areas for improvement. Such as, clarifying the contributions of the model, connecting related works to the proposed model, explaining the advantages of certain design choices, and presenting experimental parameters in a tabular format. Additionally, suggestions are made to enrich the experimental design by considering new evaluation indicators and improving language quality throughout the paper. Additionally, points are raised regarding the need for detailed data preprocessing descriptions, formula explanations, and algorithm explanations.

See the feedback for more details.

Reviewer 1 ·

Basic reporting

This paper proposes an accounting feature extraction method based on multi-modal information embedding. Double-branch structure and feature pyramid network are used to achieve feature extraction of information involved in accounting. Experiments show that the proposed method can achieve high accuracy, but there are still the following shortcomings:
1. The description of the experimental part of the abstract is too little to highlight the contribution of the model in this paper. I would suggest that the author enrich the content;
2. In the last paragraph of the introduction, the author does not highlight the contribution points of the article, here the author should redescribe, perhaps arrange the contribution points by points;
3. In the related works, the author introduced a lot of relevant accounting technical models, but did not make a connection with the model proposed in this paper;

Experimental design

4. What are the advantages of designing a parallel subnet structure after the backbone network mentioned in section 3.1?
5. The introduction of model parameters and experimental parameters in Section 4.1 can be considered in tabular form;

Validity of the findings

6. In addition to the basic evaluation indicators, we may consider using new indicators to enrich the experiment, such as confusion matrix and so on;

Additional comments

7. There are a few places where the language level is not good enough, and the author needs to check it carefully and re-polish it;
8. The content of the conclusion is highly similar to the abstract, and the content is not sufficient, and the analysis of the application methods in this field and the introduction of the prospects and shortcomings are lacking.

Reviewer 2 ·

Basic reporting

In this paper, a multi-objective parameter selection method based on parallel genetic algorithm and an evaluation method of accounting system for recurrent transformers are proposed to improve the financial situation of enterprises, and good experimental results have been obtained.

Experimental design

the following points still need to be improved:

For the comprehensive financial data set adopted in this paper, there seems to be a lack of corresponding data preprocessing process, and I would suggest the author to describe it in detail;
There are many formulas in the third and fourth parts of the paper, which describe the model method process and the experimental evaluation index respectively, but the corresponding formula explanation should be paid attention to;
A parallel genetic algorithm is proposed in section 3.2 for data initialization via binary encoding, which I would recommend the authors go into detail;
(

Validity of the findings

Compared with different models in section 4.3, the results of the model method in this paper have great advantages, but it seems that all are single models, can we consider mixed models?
Given the innovative and generalizing ability of the articles, I would suggest adding more excellent articles from excellent journals in recent years;
In the structure of the paper, it is necessary to consider adding a discussion section to focus on summarizing the model methods in this paper and discussing the comparison of the model and its application in the field and its prospects.

Additional comments

The overall abstract content should be described according to the research background, research purpose, research method, research experimental results and prospects;
The overall content of the introduction lacks logic and rigor, and the overall length is too small, the content is not full enough, and the cohesion is not very sufficient. The author can consider improving from these aspects;

---

## Round 0.2 · accepted · Accept

Both reviewers have confirmed that the authors have addressed all of their comments.

Reviewer 1 ·

Basic reporting

All changes have been completed

Experimental design

All changes have been completed

Validity of the findings

All changes have been completed

Reviewer 2 ·

Basic reporting

The revised version of the paper is acceptable for publication

Experimental design

The experiments are done with sufficient data set and are of acceptable quality

Validity of the findings

The paper proposes valid results which may be useful for the research community

Additional comments

Thanks for revision and incorporating my comments. Good luck